# Incorporating Side Information by Adaptive Convolution

**Di Kang**      **Debarun Dhar**      **Antoni B. Chan**
Department of Computer Science
City University of Hong Kong
{dkang5-c, ddhar2-c}@my.cityu.edu.hk, abchan@cityu.edu.hk

## Abstract

Computer vision tasks often have side information available that is helpful to solve the task. For example, for crowd counting, the camera perspective (e.g., camera angle and height) gives a clue about the appearance and scale of people in the scene. While side information has been shown to be useful for counting systems using traditional hand-crafted features, it has not been fully utilized in counting systems based on deep learning. In order to incorporate the available side information, we propose an adaptive convolutional neural network (ACNN), where the convolution filter weights adapt to the current scene context via the side information. In particular, we model the filter weights as a low-dimensional manifold within the high-dimensional space of filter weights. The filter weights are generated using a learned "filter manifold" sub-network, whose input is the side information. With the help of side information and adaptive weights, the ACNN can disentangle the variations related to the side information, and extract discriminative features related to the current context (e.g. camera perspective, noise level, blur kernel parameters). We demonstrate the effectiveness of ACNN incorporating side information on 3 tasks: crowd counting, corrupted digit recognition, and image deblurring. Our experiments show that ACNN improves the performance compared to a plain CNN with a similar number of parameters. Since existing crowd counting datasets do not contain ground-truth side information, we collect a new dataset with the ground-truth camera angle and height as the side information.

## 1 Introduction

Computer vision tasks often have side information available that is helpful to solve the task. Here we define "side information" as auxiliary metadata that is associated with the main input, and that affects the appearance/properties of the main input. For example, the camera angle affects the appearance of a person in an image (see Fig. 1 top). Even within the same scene, a person's appearance changes as they move along the ground-plane, due to changes in the relative angles to the camera sensor. Most deep learning methods ignore the side information, since if given enough data, a sufficiently large deep network should be able to learn internal representations that are invariant to the side information. In this paper, we explore how side information can be directly incorporated into deep networks so as to improve their effectiveness.

Our motivating application is crowd counting in images, which is challenging due to complicated backgrounds, severe occlusion, low-resolution images, perspective distortion, and different appearances caused by different camera tilt angles. Recent methods are based on crowd density estimation [1], where each pixel in the crowd density map represents the fraction of people in that location, and the crowd count is obtained by integrating over a region in the density map. The current state-of-the-art uses convolutional neural networks (CNN) to estimate the density maps [2–4]. Previous works have also shown that using side information, e.g., the scene perspective, helps to improve crowd counting accuracy [5, 6]. In particular, when extracting hand-crafted features (e.g., edge and texture statistics) [5–9] use scene perspective normalization, where a "perspective weight" is applied at each

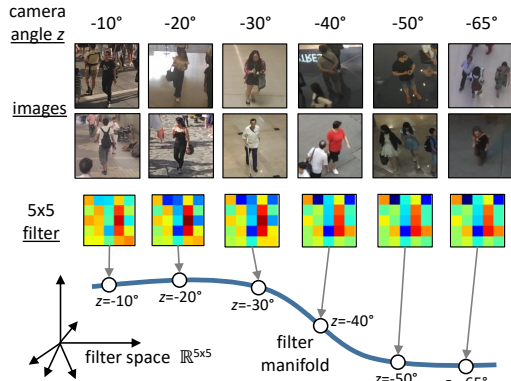

camera angle $z$    -10°    -20°    -30°    -40°    -50°    -65°

images

5x5 filter

$z$=-10°   $z$=-20°   $z$=-30°   $z$=-40°

filter space $\mathbb{R}^{5\times5}$

filter manifold

$z$=-50°   $z$=-65°

**Figure 1:** (top) changes in people's appearance due to camera angle, and the corresponding changes in a convolution filter; (bottom) the filter manifold as a function of the camera angle. Best viewed in color.

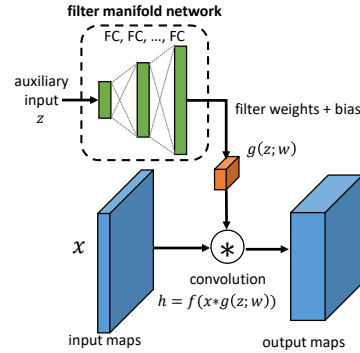

filter manifold network

FC, FC, ..., FC

auxiliary input $z$

filter weights + bias

$g(z; w)$

$x$

convolution

$h = f(x * g(z; w))$

input maps      output maps

**Figure 2:** The adaptive convolutional layer with filter manifold network (FMN). The FMN uses the auxiliary input to generate the filter weights, which are then convolved with the input maps.

pixel location during feature extraction, to adjust for the scale of the object at that location. To handle scale variations, typical CNN-based methods resize the input patch [2] based on the perspective weight, or extract features at different scales via multiple columns [3] or a pyramid of input patches [4]. However, incorporating other types of side information into the CNN is not as straightforward. As a result, all the difficulties due to various contexts, including different backgrounds, occlusion, perspective distortion and different appearances caused by different camera angles are entangled, which may introduce an extra burden on the CNNs during training. One simple solution is to add an extra image channel where each pixel holds the side information [10], which is equivalent to using 1st-layer filter bias terms that change with the side information. However, this may not be the most effective solution when the side information is a high-level property with a complex relationship with the image appearance (e.g., the camera angle).

Our solution in this paper is to disentangle the context variations explicitly in the CNN by modifying the filter weights adaptively. We propose an adaptive CNN (ACNN) that uses side information (e.g., the perspective weight) as an auxiliary input to adapt the CNN to different scene contexts (e.g., appearance changes from high/low angle perspectives, and scale changes due to distance). Specifically, we consider the filter weights in each convolutional layer as points on a low-dimensional manifold, which is modeled using a sub-network where the side information is the input and the filter weights are the outputs. The filter manifold is estimated during training, resulting in different convolution filters for each scene context, which disentangles the context variations related to the side information. In the ACNN, the convolutional layers focus only on those features most suitable for the current context specified by the side information, as compared to traditional CNNs that use a fixed set of filters over all contexts. In other words, the feature extractors are tuned for each context.

We test the effectiveness of ACNN at incorporating side information on 3 computer vision applications. First, we perform crowd counting from images using an ACNN with the camera parameters (perspective value, or camera tilt angle and height) as side information. Using the camera parameters as side information, ACNN can perform cross-scene counting without a fine-tuning stage. We collect a new dataset covering a wide range of angles and heights, containing people from different viewpoints. Second, we use ACNN for recognition of digit images that are corrupted with salt-and-pepper noise, where the noise level is the side information. Third, we apply ACNN to image deburring, where the blur kernel parameters are the side information. A single ACNN can be trained to deblur images for any setting of the kernel parameters. In contrast, using a standard CNN would require training a separate CNN for each combination of kernel parameters, which is costly if the set of parameter combinations is large. In our experiments, we show that ACNN can more effectively use the side information, as compared to traditional CNNs with similar number of parameters – moving parameters from static layers to adaptive layers yields stronger learning capability and adaptability.

The contributions of this paper are three-fold: 1) We propose a method to incorporate the side information directly into CNN by using an adaptive convolutional layer whose weights are generated via a filter manifold sub-network with side information as the input; 2) We test the efficacy of ACNN on a variety of computer vision applications, including crowd counting, corrupted digit recognition, and non-blind image deburring, and show that ACNN is more effective than traditional CNNs with

similar number of parameters. 3) We collect a new crowd counting dataset covering a wide range of viewpoints and its corresponding side information, i.e. camera tilt angle and camera height.

## 2 Related work

### 2.1 Adapting neural networks

The performance of a CNN is affected if the test set is not from the same data distribution as the training set [2]. A typical approach to adapting a CNN to new data is to select a pre-trained CNN model, e.g. AlexNet [11], VGG-net [12], or ResNet [13] trained on ImageNet, and then fine-tune the model weights for the specific task. [2] adopts a similar strategy – train the model on the whole dataset and then fine-tune using a subset of image patches that are similar to the test scene.

Another approach is to adapt the input data cube so that the extracted features and the subsequent classifier/regressor are better matched. [14] proposes a trainable "Spatial Transformer" unit that applies an image transformation to register the input image to a standard form before the convolutional layer. The functional form of the image transformation must be known, and the transformation parameters are estimated from the image. Because it operates directly on the image, [14] is limited to 2D image transformations, which work well for 2D planar surfaces in an image (e.g., text on a flat surface), but cannot handle viewpoint changes of 3D objects (e.g. people). In contrast, our ACNN changes the feature extraction layers based on the current 3D viewpoint, and does not require the geometric transformation to be known.

Most related to our work are dynamic convolution [15] and dynamic filter networks [16], which use the input image to dynamically generate the filter weights for convolution. However, their purpose for dynamically generating filters is quite different from ours. [15, 16] focus on image prediction tasks (e.g., predicting the next frame from the previous frames), and the dynamically-generated filters are mainly used to transfer a pixel value in the input image to a new position in the output image (e.g., predicting the movement of pixels between frames). These input-specific filters are suitable for low-level tasks, i.e. the input and the output are both in the same space (e.g., images). But for high-level tasks, dramatically changing features with respect to its input is not helpful for the end-goal of classification or regression. In contrast, our purpose is to include side information into supervised learning (regression and classification), by learning how the discriminative image features and corresponding filters change with respect to the side information. Hence, in our ACNN, the filter weights are generated from an auxiliary input corresponding to the side information.

HyperNetworks [17] use relaxed weight-sharing between layers/blocks, where layer weights are generated from a low-dimensional linear manifold. This can improve the expressiveness of RNNs, by changing the weights over time, or reduce the number of learnable parameters in CNNs, by sharing weight bases across layers. Specifically, for CNNs, the weight manifold of the HyperNetwork is shared across layers, and the inputs/embedding vectors of the HyperNetwork are independently learned for every layer during training. The operation of ACNNs is orthogonal to HyperNetworks - in ACNN, the weight manifold is trained independently for each layer, and the input/side information is shared across layers. In addition, our goal is to incorporate the available side information to improve the performance of the CNN models, which is not considered in [17].

Finally, one advantage of [14–17] is that no extra information or label is needed. However, this also means they cannot effectively utilize the available side information, which is common in various computer vision tasks and has been shown to be helpful for traditional hand-crafted features [5].

### 2.2 Crowd density maps

[1] proposes the concept of an object density map whose integral over any region equals to the number of objects in that region. The spatial distribution of the objects is preserved in the density map, which also makes it useful for detection [18, 19] and tracking [20]. Most of the recent state-of-the-art object counting algorithms adopt the density estimation approach [2–4, 8, 21]. CNN-based methods [2–4] show strong cross-scene prediction capability, due to the learning capacity of CNNs. Specifically, [3] uses a multi-column CNN with different receptive field sizes in order to encourage different columns to capture features at different scales (without input scaling or explicit supervision), while [4] uses a pyramid of input patches, each sent to separate sub-network, to consider multiple scales. [2] introduces an extra fine-tuning stage so that the network can be better adapted to a new scene.

In contrast to [2, 3], we propose to use the existing side information (e.g. perspective weight) as an input to adapt the convolutional layers to different scenes. With the adaptive convolutional layers,

only the discriminative features suitable for the current context are extracted. Our experiments show that moving parameters from static layers to adaptive layers yields stronger learning capability.

## 2.3 Image deconvolution

Existing works [22–24] demonstrate that CNNs can be used for image deconvolution and restoration. With non-blind deblurring, the blur kernel is known and the goal is to recover the original image. [23] concatenate a deep deconvolution CNN and a denoising CNN to perform deblurring and artifact removal. However, [23] requires a separate network to be trained for each blur kernel family and kernel parameter. [24] trains a multi-layer perceptron to denoise images corrupted by additive white Gaussian (AWG) noise. They incorporate the side information (AWG standard deviation) by simply appending it to the vectorized image patch input. In this paper, we use the kernel parameter as an auxiliary input, and train a single ACNN for a blur kernel family (for all its parameter values), rather than for each parameter separately. During prediction, the "filter-manifold network" uses the auxiliary input to generate the appropriate deblurring filters, without the need for additional training.

## 3 Adaptive CNN

In this section, we introduce the adaptive convolutional layer and the ACNN.

### 3.1 Adaptive convolutional layer

Consider a crowd image dataset containing different viewpoints of people, and we train a separate CNN to predict the density map for each viewpoint. For two similar viewpoints, we expect that the two trained CNNs have similar convolution filter weights, as a person's appearance varies gradually with the viewpoint (see Fig. 1 top). Hence, as the viewpoint changes smoothly, the convolution filters weights also change smoothly, and thus sweep a low-dimensional manifold within the high-dimensional space of filter weights (see Fig. 1 bottom).

Following this idea, we use an *adaptive convolutional layer*, where the convolution filter weights are the outputs of a separate "filter-manifold network" (FMN, see Fig. 2). In the FMN, the side information is an auxiliary input that feeds into fully-connected layers with increasing dimension (similar to the decoder stage of an auto-encoder) with the final layer outputting the convolution filter weights. The FMN output is reshaped into a 4D tensor of convolution filter weights (and bias), and convolved with the input image. Note that in contrast to the traditional convolutional layer, whose filter weights are fixed during the inference stage, the filter weights of an adaptive convolutional layer change with respect to the auxiliary input. Formally, the adaptive convolutional layer is given by $h = f(x * g(z; w))$, where $z$ is the auxiliary input, $g(\cdot; w)$ is the filter manifold network with tunable weights $w$, $x$ is the input image, and $f(\cdot)$ is the activation function.[1]

Training the adaptive convolutional layer involves updating the FMN weights $w$, thus learning the filter manifold as a function of the auxiliary input. During inference, the FMN interpolates along the filter manifold using the auxiliary input, thus adapting the filter weights of the convolutional layer to the current context. Hence adaptation does not require fine-tuning or transfer learning.

### 3.2 Adaptive CNN for crowd counting

We next introduce the ACNN for crowd counting. Density map estimation is not as high-level a task as recognition. Since the upper convolutional layers extract more abstract features, which are not that helpful according to both traditional [1, 5] and deep methods [2, 3], we will not use many convolutional layers. Fig. 3 shows our ACNN for density map estimation using two convolutional stages. The input is an image patch, while the output is the crowd density at the center of the patch. All the convolutional layers use the ReLU activation, and each convolutional layer is followed by a local response normalization layer [11] and a max pooling layer. The auxiliary input for the FMN is the perspective value for the image patch in the scene, or the camera tilt angle and camera height. For the fully-connected stage, we use multi-task learning to improve the training of the feature extractors [2, 25–27]. In particular, the main regression task predicts the crowd density value, while an auxiliary classification task predicts the number of people in the image patch.

The adaptive convolutional layer has more parameters than a standard convolutional layer with the same number of filters and the same filter spatial size – the extra parameters are in the layers of the

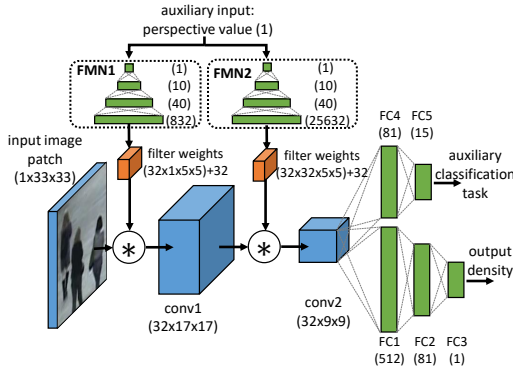

**Figure 3:** The architecture of our ACNN with adaptive convolutional layers for crowd density estimation.

| Layer | CNN | ACNN |
|---|---|---|
| FMN1 | – | 34,572 (832) |
| conv1 | 1,664 (64) | 0 (32) |
| FMN2 | – | 1,051,372 (25,632) |
| conv2 | 102,464 (64) | 0 (32) |
| FC1 | 2,654,720 (512) | 1,327,616 (512) |
| FC2 | 41,553 (81) | 41,553 (81) |
| FC3 | 82 (1) | 82 (1) |
| FC4 | 419,985 (81) | 210,033 (81) |
| FC5 | 1,312 (15) | 1,312 (15) |
| total | 3,221,780 | 2,666,540 |

**Table 1:** Comparison of number of parameters in each layer of the ACNN in Fig. 3 and an equivalent CNN. The number in parenthesis is the number of convolution filters, or the number of outputs of the FMN/fully-connected (FC) layer.

FMN. However, since the filters themselves adapt to the scene context, an ACNN can be effective with fewer feature channels (from 64 to 32), and the parameter savings can be moved to the FMN (e.g. see Table 1). Hence, if side information is available, a standard CNN can be converted into an ACNN with a similar number of parameters, but with better learning capability. We verify this property in the experiments.

Since most of the parameters of the FMN are in its last layer, the FMN has $O(LF)$ parameters, where $F$ is the number of filter parameters in the convolution layer and $L$ is the size of the last hidden layer of the FMN. Hence, for a large number of channels (e.g., 128 in, 512 out), the FMN will be extremely large. One way to handle more channels is to reduce the number of parameters in the FMN, by assuming that sub-blocks in the final weight matrix of the FMN form a manifold, which can be modeled by another FMN (i.e., an FMN-in-FMN). Here, the auxiliary inputs for the sub-block FMNs are generated from another network whose input is the original auxiliary input.

### 3.3 Adaptive CNN for image deconvolution

Our ACNN for image deconvolution is based on the deconvolution CNN proposed in [23]. The ACNN uses the kernel blur parameter (e.g., radius of the disk kernel) as the side information, and consists of three adaptive convolutional layers (see Fig. 4). The ACNN uses 12 filter channels in the first 2 layers, which yields an architecture with similar number of parameters as the standard CNN with 38 filters in [23]. The ACNN consists of two long 1D adaptive convolutional layers: twelve 121×1 vertical 1D filters, followed by twelve 1×121 horizontal 1D filters. The result is passed through a 1×1 adaptive convolutional layer to fuse all the feature maps. The input is the blurred image and the output target is the original image. We use leaky ReLU activations [28] for the first two convolutional layers, and sigmoid activation for the last layer to produce a bounded output as image. Batch normalization layers [29] are used after the convolutional layers.

During prediction, the FMN uses kernel parameter auxiliary input to generate the appropriate deblurring filters, without the need for additional training. Hence, the two advantages of using ACNN are: 1) only one network is needed for each blur kernel family, which is useful for kernels with too many parameter combinations to enumerate; 2) by interpolating along the filter manifold, ACNN can work on kernel parameters unseen in the training set.

## 4 Experiments

To show their potential, we evaluate ACNNs on three tasks: crowd counting, digit recognition with salt-and-pepper noise, and image deconvolution (deblurring). In order to make fair comparisons, we compare our ACNN with standard CNNs using traditional convolutional layers, but increase the number of filter channels in the CNN so that they have similar total number of parameters as the ACNN. We also test a CNN with side information included as an extra input channel(s) (denoted as CNN-X), where the side information is replicated in each pixel of the extra channel, as in [10].

For ACNN, each adaptive convolution layer has its own FMN, which is a standard MLP with two hidden layers and a linear output layer. The size of the FMN output layer is the same as the number of filter parameters in its associated convolution layer, and the size of the last hidden layer (e.g., 40 in Fig. 3) was selected so that the ACNN and baseline CNN have roughly equal number of parameters.

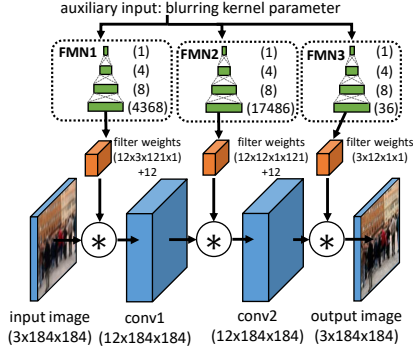

auxiliary input: blurring kernel parameter

**Figure 4:** ACNN for image deconvolution. The auxiliary input is the radius $r$ of the disk blurring kernel.

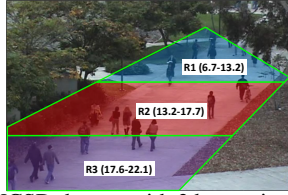

**Figure 5:** UCSD dataset with 3 bar regions. The range of perspective values are shown in parentheses.

| Method | MAE |
|---|---|
| MESA [1] | 1.70 |
| Regression forest [21] | 1.70 |
| RR [8] | 1.24 |
| CNN-patch+RR [2] | 1.70 |
| MCNN [3] | 1.32 |
| CNN | 1.26 |
| CNN-X | 1.20 |
| CNN (normalized patch) | 1.26 |
| ACNN-v1 | 1.23 |
| ACNN-v2 | 1.14 |
| ACNN-v3 | **0.96** |

**Table 2:** Comparison of mean absolute error (MAE) for counting with crowd density estimation methods on the UCSD "max" split.

| Method | R1 | R2 (unseen) | R3 | Avg. |
|---|---|---|---|---|
| CNN | 1.83 | 1.06 | 0.62 | 1.17 |
| CNN-X | 1.33 | 1.18 | 0.61 | 1.04 |
| ACNN-v1 | 1.47 | 0.95 | 0.59 | 1.00 |
| ACNN-v2 | 1.22 | **0.91** | **0.55** | **0.89** |
| ACNN-v3 | **1.15** | 1.02 | 0.63 | 0.93 |

**Table 3:** Comparison of MAE on 3 bar regions on the UCSD "max" split.

## 4.1 Crowd counting experiments

For crowd counting, we use two crowd counting datasets: the popular UCSD crowd counting dataset, and our newly collected dataset with camera tilt angle and camera height as side information.

### 4.1.1 UCSD dataset

Refer to Fig. 3 for the ACNN architecture used for the UCSD dataset. The image size is $238 \times 158$, and $33 \times 33$ patches are used. We test several variations of the ACNN: v1) only the first convolutional layer is adaptive, with 64 filters for both of the convolutional layers; v2) only the last convolutional layer is adaptive, with 64 filters for the first convolutional layer and 30 filters for its second convolutional layer; v3) all the convolutional layers are adaptive, with 32 filters for all layers, which provides maximum adaptability. The side information (auxiliary input) used for the FMN is the perspective value. For comparison, we also test a plain CNN and CNN-X with a similar architecture but using standard convolutional layers with 64 filters in each layer, and another plain CNN with input patch size normalization introduced in [2] (i.e., resizing larger patches for near-camera regions). The numbers of parameters are shown in Table 1. The count predictions in the region-of-interest (ROI) are evaluated using the mean absolute error (MAE) between the predicted count and the ground-truth.

We first use the widely adopted protocol of "max" split, which uses 160 frames (frames 601:5:1400) for training, and the remaining parts (frames 1:600, 1401:2000) for testing. The results are listed in Table 2. Our ACNN-v3, using two adaptive convolutional layers, offers maximum adaptability and has the lowest error (0.96 MAE), compared to the equivalent plain CNN and the reference methods. While CNN-X reduces the error compared to CNN, CNN-X still has larger error than ACNN. This demonstrates that the FMN of ACNN is better at incorporating the side information. In addition, using simple input patch size normalization does not improve the performance as effectively as ACNN. Examples of the learned filter manifolds are shown in Fig. 6. We also tested using 1 hidden layer in the FMN, and obtained worse errors for each version of ACNN (1.74, 1.15, and 1.20, respectively). Using only one hidden layer limits the ability to well model the filter manifold.

In the next experiment we test the effect of the side information within the same scene. The ROI of UCSD is further divided into three bar regions of the same height (see Fig. 5). The models are trained only on R1 and R3 from the training set, and tested on all three regions of the test set separately. The results are listed in Table 3. After disentangling the variations due to perspective value, the performance on R1 has been significantly improved because the ACNN uses the context information to distinguish it from the other regions. Perspective values within R2 are completely unseen during training, but our ACNN still gives a comparable or slightly better performance than CNN, which demonstrates that the FMN can smoothly interpolate along the filter manifold.

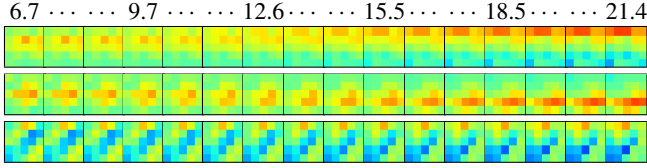

6.7 ····· 9.7 ····· 12.6 ··· ··· 15.5 ··· ··· 18.5 ··· ··· 21.4

**Figure 6:** Examples of learned filter manifolds for the 2nd convolutional layer. Each row shows one filter as a function of the auxiliary input (perspective weight), shown at the top. Both the amplitude and patterns change, which shows the adaptability of the ACNN.

| Method | MAE |
|---|---|
| LBP+RR [2, 3] | 23.97 |
| MCNN [3] | 8.80 |
| CNN | 8.72 |
| CNN-X (AH) | 9.05 |
| CNN-X (AHP) | 8.45 |
| ACNN (AH) | 8.35 |
| ACNN (AHP) | **8.00** |

**Table 4:** Counting results on CityUHK-X, the new counting dataset with side information.

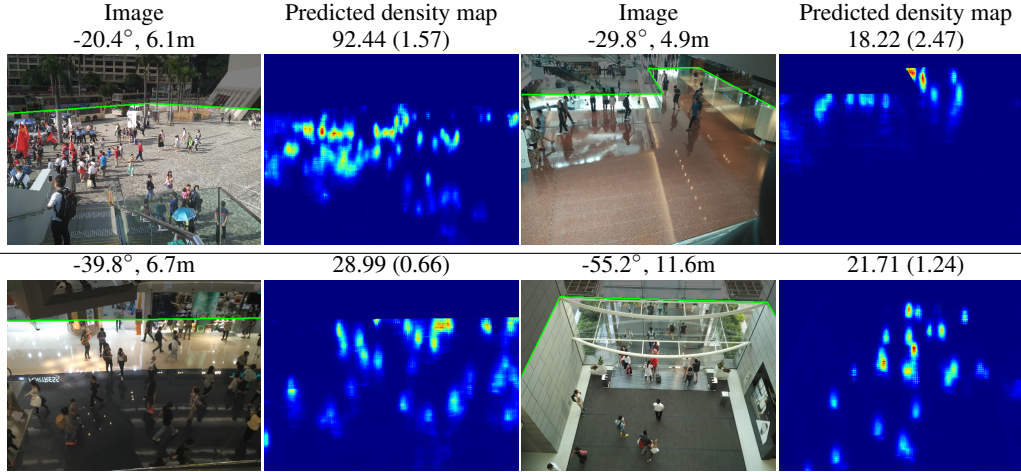

| Image<br>-20.4°, 6.1m | Predicted density map<br>92.44 (1.57) | Image<br>-29.8°, 4.9m | Predicted density map<br>18.22 (2.47) |
|---|---|---|---|
| -39.8°, 6.7m | 28.99 (0.66) | -55.2°, 11.6m | 21.71 (1.24) |

**Figure 7:** Examples of the predicted density map by our ACNN on the new CityUHK-X dataset. The extrinsic parameters and predicted count (absolute error in parenthesis) is shown above the images.

#### 4.1.2 CityUHK-X: new crowd dataset with extrinsic camera parameters

The new crowd dataset "CityUHK-X" contains 55 scenes (3,191 images in total), covering a camera tilt angle range of [-10°, -65°] and a height range of [2.2, 16.0] meters. The training set consists of 43 scenes (2,503 images; 78,592 people), and the test set comprises 12 scenes (688 images; 28,191 people). More information and demo images can be found in the supplemental. The resolution of the new dataset is 512×384, and 65×65 patches are used. The ACNN for this dataset contains three convolutional and max-pooling layers, resulting in the same output feature map size after the convolutional stage as in the ACNN for UCSD. The three adaptive convolutional layers use 40, 40 and 32 filters of size 5×5 each. The side information (auxiliary inputs) are camera tilt angle and camera height (denoted as "AH"), and the camera tilt angle, camera height, and perspective value (denoted as "AHP"). The baseline plain CNN and CNN-X use 64 filters of size 5×5 for all three convolutional layers.

Results for ACNN, the plain CNN and CNN-X, and multi-column CNN (MCNN) [3] are presented in Table 4. The plain CNN and MCNN [3], which do not use side information, obtain similar results. Using side information with ACNN decreases the MAE, compared to the plain CNN and CNN-X, with more side information improving the results (AHP vs. AH). Fig. 7 presents example results.

### 4.2 Digit recognition with salt-and-pepper noise

In this experiment, the task is to recognize handwritten digits that are corrupted with different levels of salt-and-pepper noise. The side information is the noise level. We use the MNIST handwritten digits dataset, which contains 60,000 training and 10,000 test examples. We randomly add salt-and-pepper noise (half salt and half pepper), on the MNIST images. Nine noise levels are used on the original MNIST training set from 0% to 80% with an interval of 10%, with the same number of images for each noise level, resulting in a training set of 540,000 samples. Separate validation and test sets, both containing 90,000 samples, are generated from the original MNIST test set.

We test our ACNN with the noise level as the side information, as well as the plain CNN and CNN-X. We consider two architectures: two or four convolutional layers (2-conv or 4-conv) followed by

| Architecture | No. Conv. Filters | Error Rate | No. Parameters |
|---|---|---|---|
| CNN    2-conv | 32 + 32 | 8.66% | 113,386 |
| CNN-X 2-conv | 32 + 32 | 8.49% (8.60%) | 113,674 |
| ACNN   2-conv | 32 + 26 | **7.55%** (7.64%) | 105,712 |
| CNN    4-conv | 32 + 32 + 32 + 32 | 3.58% | 131,882 |
| CNN-X 4-conv | 32 + 32 + 32 + 32 | 3.57% (3.64%) | 132,170 |
| ACNN   4-conv | 32 + 32 + 32 + 26 | **2.92%** (2.97%) | 124,208 |

**Table 5:** Digit recognition with salt-and-pepper noise, where the noise level is the side information. The number of filters for each convolutional layer and total number of parameters are listed. In the Error Rate column, the parenthesis shows the error when using the estimated side information rather than the ground-truth.

| Arch-filters | training set $r$ | $r$=3 | $r$=5 | $r$=7 | $r$=9 | $r$=11 | all | seen $r$ | unseen $r$ |
|---|---|---|---|---|---|---|---|---|---|
| blurred image | — | 23.42 | 21.90 | 20.96 | 20.28 | 19.74 | 21.26 | — | — |
| CNN [23] | {3, 7, 11} | +0.55 | -0.25 | +0.49 | +0.69 | +0.56 | +0.41 | +0.53 | +0.22 |
| CNN-X | {3, 7, 11} | +0.88 | -0.70 | +1.65 | +0.47 | +1.86 | +0.83 | **+1.46** | -0.12 |
| ACNN | {3, 7, 11} | +0.77 | +0.06 | +1.17 | +0.94 | +1.28 | **+0.84** | +1.07 | **+0.50** |
| CNN-X (blind) | {3, 7, 11} | +0.77 | -0.77 | +1.23 | +0.25 | +0.98 | +0.49 | **+0.99** | -0.26 |
| ACNN (blind) | {3, 7, 11} | +0.76 | -0.04 | +0.70 | +0.80 | +1.13 | **+0.67** | +0.86 | **+0.38** |
| CNN [23] | {3, 5, 7, 9, 11} | +0.28 | +0.45 | +0.62 | +0.86 | +0.59 | +0.56 | +0.56 | — |
| CNN-X | {3, 5, 7, 9, 11} | +0.99 | +1.38 | +1.53 | +1.60 | +1.55 | **+1.41** | **+1.41** | — |
| ACNN | {3, 5, 7, 9, 11} | +0.71 | +0.92 | +1.00 | +1.28 | +1.22 | +1.03 | +1.03 | — |
| CNN-X (blind) | {3, 5, 7, 9, 11} | +0.91 | +1.06 | +0.81 | +1.12 | +1.24 | **+1.03** | **+1.03** | — |
| ACNN (blind) | {3, 5, 7, 9, 11} | +0.66 | +0.79 | +0.64 | +1.12 | +1.04 | +0.85 | +0.85 | — |

**Table 6:** PSNRs for image deconvolution experiments. The PSNR for the blurred input image is in the first row, while the other rows are the change in PSNR relative to that of the blurred input image. Blind means the network takes estimated auxiliary value (disk radius) as the side information.

two fully-connected (FC) layers.[2] For ACNN, only the 1st convolutional layer is adaptive. All convolutional layers use $3\times3$ filters. All networks use the same configuration for the FC layers, one 128-neuron layer and one 10-neuron layer. ReLU activation is used for all layers, except the final output layer which uses soft-max. Max pooling is used after each convolutional layer for the 2-conv network, or after the 2nd and 4th convolutional layers for the 4-conv network.

The classification error rates are listed in Table 5. Generally, adding side information as extra input channel (CNN-X) decreases the error, but the benefit diminishes as the baseline performance increases – CNN-X 4-conv only decreases the error rate by 0.01% compared with CNN. Using ACNN to incorporate the side information can improve the performance more significantly. In particular, for ACNN 2-conv, the error rate decreases 0.94% (11% relatively) from 8.49% to 7.55%, while the error rate decreases 0.65% (18% relatively) from 3.57% to 2.92% for ACNN 4-conv.

We also tested the ACNN when the noise level is unknown – The noise level is estimated from the image, and then passed to the ACNN. To this end, a 4-layer CNN (2 conv. layers, 1 max-pooling layer and 2 FC layers) is trained to predict the noise level from the input image. The error rate increases slightly when using the estimated noise level (e.g., by 0.05% for the ACNN 4-conv, see Table 5). More detailed setting of the networks can be found in the supplemental.

### 4.3 Image deconvolution

In the final experiment, we use ACNN for image deconvolution (deblurring) where the kernel blur parameter is the side information. We test on the Flickr8k [31] dataset, and randomly select 5000 images for training, 1400 images for validation, and another 1600 images for testing. The images were blurred uniformly using a disk kernel, and then corrupted with additive Gaussian noise (AWG) and JPEG compression as in [23], which is the current state-of-the-art for non-blind deconvolution using deep learning. We train the models with images blurred with different sets of kernel radii $r \in \{3, 5, 7, 9, 11\}$. The test set consists of images blurred with all $r \in \{3, 5, 7, 9, 11\}$. The evaluation is based on the peak signal-to-noise ratio (PSNR) between the deconvolved image and the original image, relative to the PSNR of the blurred image.

The results are shown in Table 6 using different sets of radii for the training set. First, when trained on the full training set, ACNN almost doubles the increase in PSNR, compared to the CNN (+1.03dB vs. +0.56dB). Next, we consider a reduced training set with radii $r \in \{3, 7, 11\}$, and ACNN again doubles the increase in PSNR (+0.84dB vs. +0.41dB). The performance of ACNN on the unseen radii $r \in \{5, 9\}$ is better than CNN, which demonstrates the capability of ACNN to interpolate along

the filter manifold for unseen auxiliary inputs. Interestingly, CNN-X has higher PSNR than ACNN on seen radii, but lower PSNR on unseen radii. CNN-X cannot well handle interpolation between unseen aux inputs, which shows the advantage of explicitly modeling the filter manifold.

We also test CNN-X and ACNN for blind deconvolution, where we estimate the kernel radius using manually-crafted features and random forest regression (see supplemental). For the blind task, the PSNR drops for CNN-X (0.38 on $r \in \{3, 5, 7, 9, 11\}$ and 0.34 on $r \in \{3, 7, 11\}$) are larger than ACNN (0.18 and 0.17), which means CNN-X is more sensitive to the auxiliary input.

Example learned filters are presented in Fig. 8, and Fig. 9 presents examples of deblurred images. Deconvolved images using CNN are overly-smoothed since it treats images blurred by all the kernels uniformly. In contrast, the ACNN result has more details and higher PSNR.

On this task, CNN-X performs better than ACNN on the seen radii, most likely because the relationship between the side information (disk radius) and the main input (sharp image) is not complicated and deblurring is a low-level task. Hence, incorporating the side information directly into the filtering calculations (as an extra channel) is a viable solution[3]. In contrast, for the crowd counting and corrupted digit recognition tasks, the relationship between the side information (camera angle/height or noise level) and the main input is less straightforward and not deterministic, and hence the more complex FMN is required to properly adapt the filters. Thus, the adaptive convolutions are not universally applicable, and CNN-X could be used in some situations where there is a simple relationship between the auxiliary input and the desired filter output.

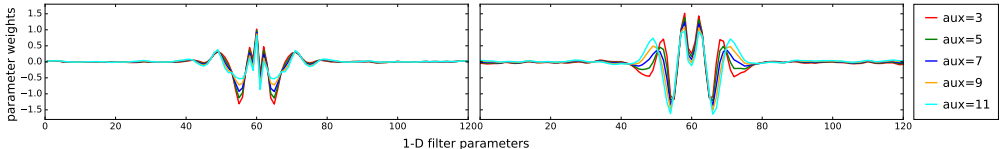

**Figure 8:** Two examples of filter manifolds for image deconvolution. The y-axis is the filter weight, and x-axis is location. The auxiliary input is the disk kernel radius. Both the amplitude and the frequency can be adapted.

(a) Original (target)  (b) Blurred (input) PSNR=24.34  (c) CNN [23] PSNR=25.30  (d) ACNN PSNR=26.04

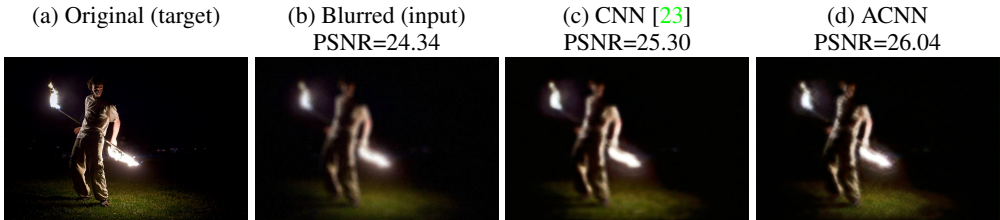

**Figure 9:** Image deconvolution example: (a) original image; (b) blurred image with disk radius of 7; deconvolved images using (c) CNN and (d) our ACNN.

## 5 Conclusion

In this paper, we propose an adaptive convolutional neural network (ACNN), which employs the available side information as an auxiliary input to adapt the convolution filter weights. The ACNN can disentangle variations related to the side information, and extract features related to the current context. We apply ACNN to three computer vision applications: crowd counting using either the camera angle/height and perspective weight as side information, corrupted digit recognition using the noise level as side information, and image deconvolution using the kernel parameter as side information. The experiments show that ACNN can better incorporate high-level side information to improve performance, as compared to using simple methods such as including the side information as an extra input channel.

The placement of the adaptive convolution layers is important, and should consider the relationship between the image content and the aux input, i.e., how the image contents changes with respect to the auxiliary input. For example, for counting, the auxiliary input indicates the amount of perspective distortion, which geometrically transforms the people's appearances, and thus adapting the 2nd layer is more helpful since changes in object configuration are reflected in mid-level features. In contrast, salt-and-pepper-noise has a low-level (local) effect on the image, and thus adapting the first layer, corresponding to low-level features, is sufficient. How to select the appropriate convolution layers for adaptation is interesting future work.

## Acknowledgments

The work described in this paper was supported by a grant from the Research Grants Council of the Hong Kong Special Administrative Region, China (Project No. [T32-101/15-R]), and by a Strategic Research Grant from City University of Hong Kong (Project No. 7004682). We gratefully acknowledge the support of NVIDIA Corporation with the donation of the Tesla K40 GPU used for this research.

## Footnotes

[1]To reduce clutter, here we do not show the bias term for the convolution.

[2] On the clean MNIST dataset, the 2-conv and 4-conv CNN architectures achieve 0.81% and 0.69% error, while the current state-of-the-art is $\sim$0.23% error [30].

[3]The extra channel is equivalent to using an adaptive bias term for each filter in the 1st convolutional layer.

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
