[Supplementary Material]

# Incorporating Side Information by Adaptive Convolution – Supplementary Material

**Di Kang**      **Debarun Dhar**      **Antoni B. Chan**
Department of Computer Science
City University of Hong Kong
{dkang5-c, ddhar2-c}@my.cityu.edu.hk, abchan@cityu.edu.hk

## 1 CityUHK-X: Crowd dataset with extrinsic camera parameters

In this section we provide additional information about the CityUHK-X dataset.

Camera extrinsic parameters, such as camera tilt angle and camera height, are useful types of side information for crowd counting. The camera tilt angle affects a person's appearance, while the distance affects the scale. Note that the camera tilt angle/height can be used to estimate the perspective map, whose values indicate the size of a person appearing at each location. As existing datasets do not contain the side information of extrinsic camera parameters, here we collect a new dataset consisting of indoor/outdoor scenes from various camera angles and heights. The scenes were captured using a smartphone camera, which was placed on a tripod to keep it stable. The camera tilt angle was recorded using the accelerometer tilt-sensor of the smartphone, and the height of the camera to the ground-plane was measured using a laser range finder. The perspective map for each scene is estimated from the camera extrinsic parameters.

### 1.1 Dataset information

CityUHK-X contains 55 scenes, covering a camera tilt angle range of [-10°, -65°] and a height range of [2.2, 16.0] meters. Each scene contains 58 images on average, with a time delay of $11 \pm 20$ seconds between images. The total number of images is 3,191, collected over $\sim$10 hours. The training set consists of 43 scenes (2,503 images; 78,592 people), and the test set comprises 12 scenes (688 images; 28,191 people). Similar to [1], the ground truth density map used for training and testing is generated using a Gaussian density with standard deviation that varies according to the perspective map ($\sigma_h = \frac{1}{5}M_p, \sigma_v = \frac{1}{2}M_p$), where $M_p$ is the perspective value at location $p$. However, instead of using two Gaussians to form a human shape (head and body), we use only a single elliptical Gaussian (for the body), since in this dataset the appearance of people changes significantly due to a larger range of camera tilt angles, compared to WorldExpo [1].

Refer to Fig. 1 for the distribution of the camera tilt angle and height of this dataset. Some demo images are in Fig. 3

### 1.2 Perspective map estimation from camera parameters

We use the camera tilt angle, camera height and the camera's vertical field of view (FOV) to estimate the perspective map of a scene without manual labeling (c.f., [1]). Following the convention in [1], the value in the perspective map is directly proportional to the size of a person appearing at the corresponding location in the image.

Since we know the camera's FOV (44.6°) and the image resolution, we can project a line through each image pixel to the scene (see Fig. 2). The appearance of a person changes with the camera tilt angle: side-view for low camera tilt angles, and top-view for high camera tilt angles. Using the scene geometry, we estimate the corresponding real-world projected depth $d$ and height $h$ for each pixel of

Figure 1: Distribution of camera tilt angle and height for scenes in CityUHK-X.

the image, and obtain an estimate of a person's projected length on that image pixel as $a = \sqrt{d \cdot h}$, in order to represent both cases of a person's side-view and top-view. The perspective value is $1/a$, which represents how many pixels are in 1 meter in the real world. Refer to Fig. 3 for the estimated perspective map, larger values indicate locations where people appear larger.

Figure 2: Geometric relationships used to estimate the perspective map for a scene.

## 2  Details about the crowd counting ACNN

During training, we have two branches, one main regression branch and one auxiliary classification branch. The loss function used for the regression branch is squared Euclidean distance,

$$\ell_{\text{density}}(d, \hat{d}) = (d - \hat{d})^2, \tag{1}$$

where $d$ is the ground truth density value and $\hat{d}$ is the predicted density value. The loss function used for the auxiliary classification task is categorical cross entropy,

$$\ell_{\text{aux}}(p, \hat{p}) = \sum_i -p_i \log \hat{p}_i \tag{2}$$

where $p_i$ is the true probability of class $i$ (i.e., 1 if the true class is $i$, and 0 otherwise), and $\hat{p}_i$ is the predicted probability of class $i$. The regression and classification tasks are combined into a single weighted loss function for training,

$$\ell = \lambda \ell_{\text{density}}(d, \hat{d}) + \ell_{\text{aux}}(p, \hat{p}), \tag{3}$$

where we use the weight value $\lambda = 100$ in our implementation. All the networks are trained using SGD with momentum with learning rate decay.

Some demo images of the predicted density maps are listed in 4.

Figure 3: Example images along with their estimated perspective maps, and ground-truth density maps from different scenes in this dataset.

| Image -29.8°, 4.9m | Predicted density map 18.22 (2.47) | Image -39.8°, 6.7m | Predicted density map 28.99 (0.66) |
| --- | --- | --- | --- |
| -50.0°, 6.7m | 13.62 (0.71) | -20.4°, 6.1m | 92.44 (1.57) |
| -14.6°, 3.6m | 29.93 (34.76) | -21.3°, 4.0m | 31.93 (18.85) |
| -35.2°, 5.3m | 33.56 (4.58) | -55.2°, 11.6m | 21.71 (1.24) |

Figure 4: Examples of the predicted density map by our ACNN. The predicted count is above, and the absolute error is in parentheses.

# 3 Details about the digit recognition ACNN

For all the digit recognition networks, they have the same configuration for their FC part, one 128-neuron layer and one 10-neuron layer. All the convolutional layers use 3×3 filters. ReLU activation is used for all layers, except the final output layer which uses soft-max. For networks with 2 convolutional layers, 2 max pooling layers are used after each convolutional layer, with pooling size of $2 \times 2$ and strides of $2 \times 2$. Valid border mode/without zero-padding is used for both convolutional layers and pooling layers. For networks with 4 convolutional layers, the extra two convolutional layers (No. 1 and 3) use same border mode/with zero-padding so that the final output feature maps after all convolutional layers remains the same size ($5 \times 5$).

The network used to prediction the noise level contains 2 convolutional layers (use $8\ 5 \times 5$ filters each), one max pooling layer (pooling size of $2 \times 2$ and strides of $2 \times 2$) and two FC layers (128 and 1 neuron respectively), 104,473 parameters in total. Valid border mode/without zero-padding is used for both convolutional layers and pooling layers. The loss function is categorical cross entropy (see Eq. 2). All networks are trained using SGD with momentum with learning rate decay.

# 4 Details about the image deconvolution ACNN

During training, the loss function used is mean squared Euclidean distance since the regression target is an image. All the networks are trained using Nadam (Adam with Nesterov momentum) [3].

In the case of blind deconvolution, the method we use to predict the disk kernel radius is to use hand-crafted features along with a random forest regressor. The 5D feature vector fed into the random forest regression is calculated for each image by averaging over 10 randomly sampled patches. The feature vector includes the variance of the Laplacian [4], saturation [5] and 3 features calculated from radially averaged power spectrum [5, 6]. Specifically, a line is used to fit the power spectrum curve in the loglog scale. The slope and intercept of the fitting line, and the residual (MAE between the fitting line and the power spectrum curve) are used.

Figs. 5 and 6 present more results on our image deconvolution experiment trained on images blurred with kernel radii $r \in \{3, 5, 7, 9, 11\}$.

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

| Original image | (radius) Convolved image (r=3) PSNR=24.45 | ACNN result PSNR=25.23 | CNN result PSNR=24.55 |
|---|---|---|---|
| | (r=5) PSNR=22.57 | PSNR=23.68 | PSNR=23.07 |
| | (r=7) PSNR=21.36 | PSNR=22.63 | PSNR=21.97 |
| | (r9) PSNR=20.54 | PSNR=21.99 | PSNR=21.62 |
| | (r=11) PSNR=19.95 | PSNR=21.27 | PSNR=20.68 |

Figure 5: Image deconvolution results (part 1). Each row shows a comparison between the deconvolved images by our ACNN and normal CNN. $r$ is the blur kernel radius used to blur the image.

| Original image | (radius) Convolved image<br>(r=3) PSNR=22.86 | ACNN result<br>PSNR=23.73 | CNN result<br>PSNR= 23.29 |
| --- | --- | --- | --- |

| | (r=5) PSNR=21.15 | PSNR=22.07 | PSNR=21.59 |
| --- | --- | --- | --- |

| | (r=7) PSNR=20.14 | PSNR=21.17 | PSNR=20.61 |
| --- | --- | --- | --- |

| | (r=9) PSNR=19.49 | PSNR=20.72 | PSNR=20.20 |
| --- | --- | --- | --- |

| | (r=11) PSNR=19.01 | PSNR=20.07 | PSNR=19.47 |
| --- | --- | --- | --- |

Figure 6: Image deconvolution results (part 2). Each row shows a comparison between the deconvolved images by our ACNN and normal CNN. $r$ is the blur kernel radius used to blur the image.