[Reviews · NeurIPS 2017]

Reviewer 1



This paper proposes adapting CNNs to specific subtasks through the use of side networks that generate filters for the primary network. In particular, a side network takes as input a small number of variables (for example, camera angle), and produces as output filter parameters that the main network will use when processing an input image to solve a particular task (for example, crowd counting). The paper demonstrates use of such adaptive side layers to parametrize multiple CNN layers. Strengths of this paper are mixed. In terms of novelty, one might view the proposed design as a specific instantiation of hypernetworks [17]; the paper acknowledges this connection. Experiments seem to be well-designed and actually establish the utility of the proposed side network design in terms of both performance and efficiency. In particular, the side network design outperforms baseline methods which only append the side knowledge to the input. The side network designs are also able to achieve such performance with fewer parameters than the baseline networks. Experiments demonstrate utility across multiple tasks.

Reviewer 2



The paper proposes a new method for incorporating side information into Convolutional Neural Networks (CNNs) through adaptive convolution. The idea is to provide an adaptive convolutional layer with filter manifold network (FMN) that uses the auxiliary input to generatethe filter weights that are convolved with the input maps. The paper performs experiments on several tasks where data has undergone manipulation and show that the adaptive convolution provides better effectiveness that baseline CNN or my incorporating side information as an extra channel. The strength of the paper is that the work is well motivated by an application of people counting in video, where the cameras provide a wide range of perspectives, viewpoints and scales in capturing the real world scenes. The paper shows that the adaptive convolution method improves accuracy by incorporating side information that modifies the filter weights. The weakness of the paper is that other experiments not sufficiently grounded in real data with real corruption requiring the incorporation of side information. Instead, the authors have manipulated data sets by adding salt and pepper noise or by blurring. On one hand these manipulations are too simple, and they do not reflect the extent to which images may be corrupted in real applications. The paper will be improved by extending the experiments to cover genuine scenarios with real data. Since the problem is sufficiently motivation by needs in practice, it should be possible to use real data and scenarios that allow side information.

Reviewer 3



Summary of the Paper: This work proposes to use adaptive convolutions (also called 'cross convolutions') to incorporate side information (e.g., camera angle) into CNN architectures for vision tasks (e.g., crowd counting). The filter weights in each adaptive convolution layer are predicted using a separate neural network (one network for each set of filter weights) with is a multi-layer perceptron. This network is referred to as 'Filter Manifold Network' which takes the auxiliary side information as input and predicts the filter weights. Experiments on three vision tasks of crowd counting, digit recognition and image deconvolution indicate the potential of the proposed technique for incorporating auxiliary information. In addition, this paper contributes a new dataset for crowd counting with different camera heights and angles. Paper Strengths: - A simple yet working technique for incorporating auxiliary information into CNNs with adaptive convolutions. - With this technique, a single network is sufficient for different dataset settings (for instance, denoising with different blur kernel parameters) and authors demonstrated generalization capability of the proposed technique to interpolate between auxiliary information. - Experiments on 3 different vision tasks with good results compared to similar CNN architectures with non-adaptive convolutions. - A new dataset on crowd counting. Weaknesses: - There is almost no discussion or analysis on the 'filter manifold network' (FMN) which forms the main part of the technique. Did authors experiment with any other architectures for FMN? How does the adaptive convolutions scale with the number of filter parameters? It seems that in all the experiments, the number of input and output channels is small (around 32). Can FMN scale reasonably well when the number of filter parameters is huge (say, 128 to 512 input and output channels which is common to many CNN architectures)? - From the experimental results, it seems that replacing normal convolutions with adaptive convolutions in not always a good. In Table-3, ACNN-v3 (all adaptive convolutions) performed worse that ACNN-v2 (adaptive convolutions only in the last layer). So, it seems that the placement of adaptive convolutions is important, but there is no analysis or comments on this aspect of the technique. - The improvements on image deconvolution is minimal with CNN-X working better than ACNN when all the dataset is considered. This shows that the adaptive convolutions are not universally applicable when the side information is available. Also, there are no comparisons with state-of-the-art network architectures for digit recognition and image deconvolution. Suggestions: - It would be good to move some visual results from supplementary to the main paper. In the main paper, there is almost no visual results on crowd density estimation which forms the main experiment of the paper. At present, there are 3 different figures for illustrating the proposed network architecture. Probably, authors can condense it to two and make use of that space for some visual results. - It would be great if authors can address some of the above weaknesses in the revision to make this a good paper. Review Summary: - Despite some drawbacks in terms of experimental analysis and the general applicability of the proposed technique, the paper has several experiments and insights that would be interesting to the community. ------------------ After the Rebuttal: ------------------ My concern with this paper is insufficient analysis of 'filter manifold network' architecture and the placement of adaptive convolutions in a given CNN. Authors partially addressed these points in their rebuttal while promising to add the discussion into a revised version and deferring some other parts to future work. With the expectation that authors would revise the paper and also since other reviewers are fairly positive about this work, I recommend this paper for acceptance.